# Ni-Doped In_2_O_3_ Nanoparticles and Their Composite with rGO for Efficient Degradation of Organic Pollutants in Wastewater under Visible Light Irradiation

**DOI:** 10.3390/ijms24097950

**Published:** 2023-04-27

**Authors:** Amir Habib, Muhammad Shahzeb Khan, Muhammad Zubair, Iftikhar Ul Hasan

**Affiliations:** 1Department of Physics, College of Science, University of Hafr Al Batin, P.O. Box 1803, Hafr Al Batin 39524, Saudi Arabia; mzubair@uhb.edu.sa (M.Z.); iftikharuh@uhb.edu.sa (I.U.H.); 2Department of Mechanical Engineering, College of Engineering, University of Hafr Al Batin, P.O. Box 1803, Hafr Al Batin 39524, Saudi Arabia; mshahzeb@uhb.edu.sa

**Keywords:** indium oxide, Ni-In_2_O_3_ graphene oxide composite, photocatalysis, degradation of organic effluent

## Abstract

The efficient degradation of organic effluent is always desirable when using advanced photocatalysts with enhanced activity under visible light. Nickel-doped indium oxide (Ni-In_2_O_3_) is synthesized via a hydrothermal route as well as its composites with reduced graphene oxide (rGO). Facile synthesis and composite formation methods lead to a well-defined morphology of fabricated nanocomposite at low temperatures. The bandgap energy of indium oxide lies in the range of 3.00–4.30 eV. Its high light absorption capacity, high stability, and non-toxicity make it a choice as a photocatalyst that is active under visible light. The transition metal Ni-doping changes the indium oxide’s chemical, optical, and physicochemical properties. The Ni-In_2_O_3_ and rGO composites improved the charge transport and reduced the charge recombination. The phase analysis of the developed photocatalysts was performed using X-ray diffraction (XRD), and the morphological and structural properties were observed using advanced microscopic techniques (SEM and TEM), while UV-vis and FTIR spectroscopic techniques were used to confirm the structure and optical and chemical properties. The electrochemical properties of the photocatalysts were investigated using cyclic voltammetry (CV), linear sweep voltammetry (LSV), and electrochemical impedance spectroscopy (EIS), and the charge-transfer properties of the obtained photocatalysts and the mechanism of the photocatalytic degradation mechanism of methylene blue, a common dye used in the dyeing industry, were determined.

## 1. Introduction

Increasing environmental organic pollution has prompted researchers to find remedial measures for treating such pollution. Physical adsorption, biodegradation, oxidation, and photocatalysis are a few. Photocatalysis is used in the textile industry to degrade organic dyes in industrial wastewater [1]. It is a process in which a photocatalytic material is mixed with a dye effluent solution and irradiated with a suitable light source that may be ultraviolet (UV), visible light, or both. The photocatalyst reacts with the dye effluent and produces free radicals (e.g., superoxide/hydroxyl radicals) [2]. These free radicals help with dye degradation and break the bonds of the organic substances, converting them to less harmful or safe products [3]. A suitable photocatalytic material must possess efficient light-absorbability, non-toxicity, high photostability, high carrier mobility, and a low charge recombination rate for better photodegradation performance. The fine-tuned bandgap of such a catalyst is necessary to degrade dyes under visible light; in large bandgap materials, such as TiO_2_-activation under UV light, it limits its use to a smaller (nearly 4%) solar spectrum portion. Active photocatalyst materials have naturally caused a shift in research interest toward visible light-based materials (45% of the solar spectrum) [4,5]. Efficiency and light absorption capabilities are the other important characteristics of a photocatalyst. A perusal of highly efficient visible-light active photocatalyst materials led to using semiconductor metal-oxide nanoparticles, such as TiO_2_, ZnO, V_2_O_5_, CeO_2_, and SnO_2_ [6,7,8,9,10]. The use of these nanoparticles results in the perturbation of the semiconductor band structure due to carrier confinement, with an increase in the effective bandgap occurring. The redox potential of the photogenerated carriers is greatly enhanced in such quantized semiconductor particles; reduction reactions that cannot occur in bulk materials can occur in sufficiently small particles [11]. One technique to improve the photocatalytic activity of metal oxide photocatalysts is to dope them with transition metal ions. Recently, the doping of ZnO with Al^3+^ led to a lowering of the bandgap of the composites and improved the surface properties, which contributed to a lower charge-transfer resistance, efficient charge-carrier separation and transfer, and slower electron-hole recombination [12].

The shortcomings, such as the limited surface area of MoO_3_ as a photocatalyst, have been overcome with the incorporation of 2D matrices for better absorption in UV light, which led to a lesser photogenerated charge recombination time and decreased photocatalytic reaction kinetics [13,14]. Further investigation is required due to the fast electron-hole recombination rate in most metal oxides. Composites, such as In_2_O_3_–ZnO, for the improved degradation of methylene blue are reported in the literature [15,16]. In other instances, Sn-doped In_2_O_3_ (ITO) nanocrystals, co-catalyzed with graphitic carbon (g-C_3_N_4_) nanosheets, were reported as photocatalysts for the reduced recombination of photoexcited electron-hole pairs and improved photoelectrochemical activity for a reduction of water to H_2_ [17]. Various transition metals were doped in In_2_O_3_ by Shanmuganathan et al., where they observed an improved photocatalytic degradation of MB dye with nickel doping and attributed it to the highest photoluminescence decay lifetime of Ni-doped In_2_O_3_ [18]. Li et al. investigated the photocatalytic decomposition of perfluorooctanoic acid in the presence of In_2_O_3_-wrapped graphene [19]. Ni-doped In_2_O_3_ was synthesized using the one-step solvothermal method by Sun et al. [20] and by a modified hydrothermal technique [21]. To the best of our knowledge, no report on the degradation of MB in the presence of the reduced graphene oxide (rGO) composite of Ni-doped In_2_O_3_ is reported in the literature. 

Farjami et al. observed that α-Ni(OH)_2_ and graphene oxide alone have little catalytic activity for the oxygen reduction reaction; the Ni(OH)_2_/graphene oxide material shows significant electrocatalytic activity for the oxygen reduction reaction in alkaline solution and suggested a further investigation into this composite’s use along with other transition metals. The behavior of this Ni(OH)_2_/graphene oxide catalyst provides the impetus for further investigation of its use as an oxygen-reduction reaction catalyst, either alone or in combination with other transition metals [22]. Liu et al. have synthesized BiVO_4_/BiOBr-Ov photocatalysts containing oxygen vacancies and reported that the existence of the oxygen vacancies led to altering the photogenerated carrier transfer path [23]. Similarly, the WO_3_/BiOBr S-scheme heterojunction and CoAl-LDH/BiOBr Z-scheme photocatalyst were synthesized, and their synergistic effect with peroxymonosulfate (PMS) activation on the degradation of pollutants was studied in the recent literature [24,25]. 

## 2. Results and Discussion

### 2.1. X-ray Diffraction (XRD) Analysis

The structural and phase analyses of powdered pristine In_2_O_3_, Ni-In_2_O_3_, and rGO/Ni-In_2_O_3_ were performed using the X-ray diffraction (XRD) technique, as shown in Figure 1a. The distinctive peaks of In_2_O_3_ are positioned at 2θ = 30.52°, 36.50°, 51.89°, and 61.78°, corresponding to the (211), (222), (400), and (440) planes by matching them with the standard XRD pattern of In_2_O_3_ (JCPDS card number 06-0416) and then indexing them to the cubic bixbyite crystal structure of indium oxide. The doping of nickel in indium oxide enhances the sharpness of the peaks, which can be construed as an improvement in the material’s crystallinity. Figure 1b shows enlarged XRD patterns of the Ni-doped In_2_O_3_ powders in the 2θ range of 29.6–31.6°. The main peak (2 2 2) is clearly seen shifted towards a lower angle upon Ni-doping in Ni-doped In_2_O_3_ and rGO Ni-In_2_O_3_. This shift is attributed to the interstitial doping of Ni in the In_2_O_3_ lattice [26,27]. The famous rGO (002) peak is observable as a hump in the rGO Ni-In_2_O_3_ sample in Figure 1a [28]. In addition, the average crystallite sizes of the calcined In_2_O_3_ and Ni-In_2_O_3_ samples are calculated using the Debye–Scherrer formula and found to be 21 nm and 19 nm, respectively. The decrease in crystallite size may be attributed to the difference between the host metal ion In^3+^ (80 pm) and doped nickel metal ion Ni^2+^ (69 pm).

The XRD pattern in the figure shows that all their diffraction peaks agree well with JCPDS card no 06-0416, i.e., the cubic bixbyite-type In_2_O_3_, well-oriented along the (2 2 2) direction, as observed by Parast and Morsali et al. [29] Yang et al. [30] and Seetha et al. [21]. Other impurities’ diffraction peaks, such as NiO or Ni(OH)_2_, are not detected. The sample’s crystallinity improvement is observed with sharpness in the peak for Ni-doped In_2_O_3_. No new peak was observed, even upon the 10 wt.% and 15 wt.% Ni-doped indium oxide thin film samples, as reported by Narmada et al. [31]. It was suggested that as the ionic radius of Ni^2+^ (0.83 Å) is smaller than In^3+^ (0.94 Å), Ni^2+^ might be substituted into lattice sites. The average crystallite size of the calcined In_2_O_3_ and Ni-In_2_O_3_ samples is calculated using the Scherrer formula and found to be 21 nm and 19 nm, respectively. All the peaks corresponded with a cubic phase of the In_2_O_3_ thin films deposited at 400 °C. The data values on increasing doping concentrations show a consistent decrease in the crystalline size and lattice parameters.

### 2.2. Fourier-Transform Spectroscopy (FTIR) Analysis

Fourier-transform IR spectroscopy reveals a material’s functional groups, chemical composition, and bending or stretching vibrations. Figure 1c illustrates the FTIR pattern (a wavenumber (cm^−1^) in the range of 400–1000 cm^−1^) using a solid KBr pellet method for the pure In_2_O_3_, Ni-In_2_O_3_, and rGO/Ni-In_2_O_3_ photocatalysts. In the FTIR spectrum of pure indium oxide (In_2_O_3_), there appear few observable characteristic peaks positioned at the wavenumbers 749, 601, 565, 540, and 418 cm^−1^, and corresponds to In-coordinated oxygen (In-O), indium-to-indium stretching (In-In) and the stretching manner of the two atoms of indium mutually coordinated with oxygen (In-O-In) [32,33]. The doping of nickel with In_2_O_3_ in the presence of rGO does not change the vibrations bands, and no infrared peak associated with Ni is observed in the FTIR, which is in accordance with the XRD pattern.

### 2.3. Scanning Electron Microscopy (SEM) and EDS Analysis of rGO/Ni-In_2_O_3_

The morphological studies of pure In_2_O_3_, Ni-In_2_O_3_, and rGO/Ni-In_2_O_3_ were carried out using a scanning electron microscope (SEM), and the micrographs are presented in Figure 2. The micrographs display the details about the synthesized materials’ size, symmetry, and morphology. In Figure 2a, submicron-sized homogenous-sized particles of pure In_2_O_3_ are readily observable. The In_2_O_3_ nanoparticles are observable as agglomerates due to their small size and high surface energies. Figure 2b, as well, indicates agglomerated nickel-doped In_2_O_3_ nanoparticles. With rGO, the nanoparticles appear to have a better dispersion, indicating the linking of nanoparticles to the carbon matrix, as in Figure 2c. Defects on the surface of rGO might have attached to the OH groups on the surface of Ni- In_2_O_3_. A blend of nanoparticles residing on the rGO is evident, and aggregation of the particles, especially in Figure 2b, confirms the submicron nature of the particles due to a large surface affinity.

Nevertheless, Figure 2c,d validates the successful synthesis of nickel-doped indium-oxide nanocomposite upon rGO nanosheets, which will go through further photoelectrochemical processing (PEC) characterization. Moreover, Figure 2d shows the result of energy dispersive spectroscopy of the nickel-doped indium oxide nanoparticles congruently wrapped on reduced graphene oxide nanosheets. It confirms the presence of elements, i.e., indium (In), nickel (Ni), oxygen (O_2_), and carbon (C).

### 2.4. Transmission Electron Microscopy (TEM)

The bright- and dark-field TEM images of rGO/Ni-In_2_O_3_ nanocomposite are shown in Figure 3a–d. Uniformly dispersed Ni-In_2_O_3_ nanoparticles on the conducting platform of the rGO nanosized sheets can be seen in Figure 3a. Figure 3b,c displays higher magnification bright-field (BF-TEM) micrographs of the rGO/Ni-In_2_O_3_ photocatalyst. It confirms the crystalline nature of the synthesized Ni-In_2_O_3_ nanocrystals. The orientation of nanoparticles in the different orientations of lattices in different grains is observable. The cubic to spherical-shaped particle dimensions range from 18–20 nm. In addition, Figure 3d presents the lattice spacing of 0.9 to 1.2, which also agrees with the lattice constant of indium oxide (a = 1.0117 nm). Furthermore, Figure 3d depicts the high-resolution dark-field selected area diffraction pattern (DF-SAED) of the rGO/Ni-In_2_O_3_ photocatalyst. The spotty ring pattern of indium oxide can be found in the figure, and the concentric rings conform with the major XRD (hkl) planes (211), (222), (400), and (440) in further correspondence with the JCPDS card number 06-0416.

### 2.5. Electrical Conductivity Measurements (I–V Analysis)

The synthesized materials’ current and voltage (I–V) characteristics were conducted by forming the pellets of solid-state materials using a hydraulic press at the appropriate pressure and were further subjected to a Keithley digital picoammeter for the conductivity measurements. The potential window of −5 V to +5 V was set for estimating the effects of the doping of Ni and rGO on the electrical conductivity of In_2_O_3_ at an ambient temperature. Figure 4a presents the prepared materials’ current-to-voltage trends. The electrical conductivity profile of pure In_2_O_3_, Ni-In_2_O_3_, and rGO/Ni-In_2_O_3_ nanocomposites show linear behavior. An enhanced conducting response was observed with the doping of Ni in In_2_O_3,_ and a further rise in conductivity was observed with the incorporation of rGO nanosheets. The electric conductivity (σ) of all the materials was determined using the given Equation (1) [34].
(1)$$\sigma \;=\frac{d}{\mathrm{RA}}$$

In Equation (1), d, R, and A are the prepared pellets’ thickness, resistance, and area, respectively. The resistance of the pure In_2_O_3_, Ni-In_2_O_3_, and rGO/Ni-In_2_O_3_ nanocomposite samples was determined using the slope of their respective I–V curve. The calculated electrical conductivity (σ) of pure In_2_O_3_, Ni-In_2_O_3_, and rGO/Ni-In_2_O_3_ nanocomposites were 0.116 S/m, 7.094 S/m, and 10.811 S/m, respectively, as shown in Figure 4b. Here, an increase in the conductivity with Ni doping by 63 times and rGO addition by 93 times to pure In_2_O_3_ is noticeable and is attributed to the bandgap reduction and availability of excess active sites due to the Ni-doping and rGO presence. 

The UV-vis adsorption results augment the improved visible light absorbance by the rGO/Ni/In_2_O_3_ photocatalyst due to the Ni doping and rGO incorporation. Figure 4c–e shows the UV-vis absorption spectra of the pure In_2_O_3_, Ni-doped In_2_O_3,_ and rGO/Ni-In_2_O_3_. It can be seen that the absorbance occurs at wavelengths less than 390 nm. However, the cut-off absorption wavelength of In_2_O_3_ particles is about 470 nm. Compared with the In_2_O_3_ nanoparticles, the UV-vis spectra of rGO/Ni-In_2_O_3_ showed distinctly higher absorptions in the UV-vis region. The bandgap calculations, using Tauc plot ((αhv)2 vs. hv) in the inset of Figure 4c–e, show that a decrease in the bandgap with Ni-doping might be due to a decrease in the crystallite size. No significant effect on the bandgap was observed in the Ni-doped In_2_O_3_ composite with the rGO [11,16]. An overall decrease in the bandgap energy of the photocatalyst is conducive to the transition of electrons among energy levels, i.e., the conduction band edge (E_cb_) and valance band edge (E_vb_). It leads to an improved photocatalytic response upon visible light exposure [35].

### 2.6. Electrochemical Measurements

The electrical properties of our photocatalyst were analyzed using various supportive electrochemical measurements, such as cyclic voltammetry (CV), linear sweep voltammetry (LSV), transient chronoamperometry, and electrical impedance spectroscopy (EIS analysis). The CV study was used to obtain the redox behavior of the photocatalyst samples. The CV curve of Ni-In_2_O_3_ exhibits a shift to a positive potential and a sharp rise in the reduction current with rGO presence, which signifies the higher catalytic activity of the prepared composite (Figure 5a). The difference can be observed in linear scan voltammetry (Figure 5b). These observations indicate an electronic intermediate step introduction with Ni-doping and an overlap between the electronic levels of Ni-In_2_O_3_ and rGO [16,18]. The CV curve of In_2_O_3_ shows peaks at the potentials −0.96 V vs. Ag/AgCl reference electrode (Figure 5a). It shows mainly an anodic peak because In_2_O_3_ is an n-type semiconductor. Therefore, the anodic potential of In_2_O_3_ in the negative direction is relatively large compared to the cathodic current. 

Transient chronoamperometry is used to evaluate the charge-transfer properties of the samples within the valance and conduction bands. The photocurrent shows the ability of a metal oxide semiconductor to generate photoexcited charge carriers. The amount of photocurrent is a direct measure of the molecule splitting rate; hence, it represents the number of charge carriers generated by light irradiation. The oxidation reaction occurs on the working electrode and the photoanode, and reduction occurs on the counter electrode [36]. Figure 5c shows the transitory photocurrent response of our samples obtained by multiple on–off light cycles. All samples showed a steady response upon light irradiation, i.e., electron-hole pair generation as a photocurrent. It is observable in Figure 5c that the photo response, i.e., the photocurrent intensity of rGO/Ni-In_2_O_3,_ is the greatest of the three samples. It is concluded from the transient photo study that our rGO-based Ni-In_2_O_3_ composite has a quick charge transfer and effective charge separation ability over its counterparts, as confirmed with electrochemical impedance spectroscopy. Photocurrent response curves under the light switching on–off are used to evaluate the electronic interaction. The pure In_2_O_3_ shows a relatively low photocurrent response due to the fast recombination of the hole-electron pair. Ni-doped and rGO/Ni-In_2_O_3_ reveal that the separation rate of photogenerated electron-hole pairs has increased due to the intermediate fermi level introduction due to Ni-doping and a readily available carbon matrix as rGO. The photocurrent density for rGO/Ni-In_2_O_3_ is four to five-fold higher than that of pure In_2_O_3_, as shown in Figure 5c. 

To analyze the electron transfer properties of our photocatalyst, electrochemical impedance analysis in 0.5 M of Na_2_SO_4_ electrolyte solution, in the frequency range of 30 KHz to 5 Hz, was conducted. The EIS study evaluates the photocatalyst-electrolyte surface-charge recombination and transfer process. The Nyquist plot or the EIS spectra of the pure In_2_O_3_, Ni-In_2_O_3,_ and rGO/Ni-In_2_O_3_ samples are shown in Figure 5d. A Nyquist plot is a graph between imaginary impedance (Im(Z)) and real impedance (Re(z)). The diameter of the semi-circle shows the charge-transfer resistance R_ct_ between the electrolyte solution and photocatalyst surface [37]. It is observable that the rGO/Ni-In_2_O_3_ follows a semi-circular arc of radius smaller than the semi-circular arcs of pure and doped In_2_O_3_, indicating lower values of R_ct_. Low values of R_ct_ are an indication of improved charge-transfer behavior. This enhanced charge transfer resulted in an accelerated charge transfer to the photocatalyst surface and reduced the possibility of recombination of photogenerated electron-hole pairs [38]. The R_ct_ value of Ni-In_2_O_3_ is relatively lower than that of pure In_2_O_3_ samples. It shows that the Ni-doping of In_2_O_3_ and rGO incorporation has improved the charge-transfer property of the photocatalyst. As a result, a photocatalyst with improved organic effluent degradation ability is achieved and is further confirmed by the degradation study of MB dye [39]. 

Mott–Schottky measurements of pure In_2_O_3_, Ni-In_2_O_3,_ and the rGO/Ni-In_2_O_3_ agree well with the UV-vis spectroscopy observance of the bandgap structure calculation using the Tauc plot. A redshift is observed in the Tauc plot for Ni-In_2_O_3_ samples compared to pure In_2_O_3_. The bandgaps for pure In_2_O_3_ and the Ni-In_2_O_3_ were determined to be 3.26 and 2.57 eV, respectively. The Mott–Schottky plots with positive slopes indicate the nature of an n-type semiconductor for the samples (Figure 5e) [40]. Through the extrapolation of the Mott–Schottky curves, the E_fb_ of pure In_2_O_3_ the Ni-In_2_O_3_ is determined to be −2.1 and −0.1 and 0.09 eV vs. the reference electrode, respectively. We can see that the E_g_ and band edges of In_2_O_3_ are changed with Ni-doping, and the Fermi energy (E_F_) level is elevated due to Ni-doping [41]. According to classical semiconductor physics, the n-type characteristic is related to the position of E_F_ between the conduction band edge (E_cb_) max and the valance band edge (E_vb_). The smaller the distance between E_F_ and E_cb_, the greater the n-type characteristic. With Ni-doping, the E_cb_ is slightly reduced. At the same time, the E_F_ is lifted towards E_cb_, resulting in a smaller distance between them and, thus, enhancing the n-type characteristics, which benefit the charge-transfer kinetics at the electrode/electrolyte interface process [42]. 

### 2.7. Photocatalytic Degradation Study

#### 2.7.1. Photocatalytic Degradation Study of MB

The photocatalytic activity of pure In_2_O_3_, Ni-doped In_2_O_3_, and the rGO/Ni-In_2_O_3_ samples was studied by their efficacy in degrading the MB (5 ppm) dye solution under visible light, i.e., under 1 Sun with a solar simulator. After a suitable adsorption time in the dark, the dye solution was irradiated under visible light. The sample was collected initially after 10 min and then at the rate of 15 min each until 100 min from the reactor and subjected to UV-vis spectroscopic analysis to observe the degradation rate. 

The rGO/Ni-In_2_O_3_ photocatalyst (Figure 6b) shows a decrease in the absorption intensity after 115 min under visible light irradiation. It demonstrates better dye removal properties than pure and Ni-doped In_2_O_3_ (Figure 6c,d). Furthermore, Figure 6a shows that rGO/Ni-In_2_O_3_ degraded 95% of MB in 115 min, while pure In_2_O_3_ and Ni-doped In_2_O_3_ degraded only 40% and 55% of the MB dye, respectively. The IV measurement shows an increase in the rGO/Ni-In_2_O_3_ photocatalyst’s conductivity compared to pure and Ni-doped In_2_O_3_ (Figure 6a). The doping of In_2_O_3_ with nickel enhanced the surface deformities of In_2_O_3_. The resulting Ni-doped In_2_O_3_ has less recombination of the electron-hole pairs compared to pure In_2_O_3_, improving the photocatalytic performance observed in the doped samples. Here, Ni may act as a doping agent that inhibits the electron-hole pair recombination [18]. The fine-tuning of the bandgap of our synthesized photocatalyst due to doping, as observed in the Tauc plot and Mott–Schottky plots, affects the enhancement of electron-hole pair excitation. 

Furthermore, no shift in the maximum absorbance wavelength peak of dye in the visible region (~663 nm) is observed from the absorbance spectra of rGO/Ni-In_2_O_3_ (Figure 6b). On the other hand, the absorbance peaks of the MB dye at 663 nm and 292 nm began decreasing over time, showing the photocatalytic activity of our catalyst in both the visible and UV ranges. This decrease suggests the decomposition/mineralization of the benzene ring of the MB dye molecule [14,18].

#### 2.7.2. Kinetic Study of the Photocatalyst

Figure 7 shows the dye removal efficacy of synthesized pure In_2_O_3_, Ni-doped In_2_O_3_, and the rGO/Ni-In_2_O_3_ photocatalysts. Their efficiency (C_t_/C_0_) was evaluated through the kinetic analysis of MB dye photodegradation with time in minutes.

Figure 7 shows that the dye degradation due to the rGO/Ni-In_2_O_3_ sample is faster than pure and doped indium oxide. This is because the dye degradation follows the pseudo-first-order kinetic model, as described by Equations (2) and (3) below [16].
(2)$$C_{t}=C_{0}e^{-\mathrm{kt}}$$
(3)$$-\ln\frac{C_{t}}{C_{0}}=\mathrm{kt}$$

Here in Equations (2) and (3) C_0_ and C_t_ are the dye concentration times at time zero, and ‘t’ and k are rate constants. Figure 7b shows the graph of −ln C_t_/C_0_ versus time for the three photocatalysts. The rate constants were calculated by linear fitting of the data and are presented in the bar graph in Figure 7c. The rate-constant result was minimum for pure In_2_O_3_ (0.0018 min^−1^), medium for Ni-In_2_O_3_ (0.0028 min^−1^), and maximum for the rGO/Ni-In_2_O_3_ (0.0069 min^−1^)-based photocatalyst. The rGO/Ni-In_2_O_3_ photocatalyst removal rate is about 3.83 times faster than pure In_2_O_3_. With exposure to sunlight, the reaction started over the In_2_O_3_ surface. It supplied energy to the electrons to jump to the conduction band from the valence band, thus generating a hole in the valance band. The photogenerated electron on the surface of In_2_O_3_ reacts with the electron-accepting molecules (O_2_) already absorbed on the photocatalyst surface. It leads to the formation of O_2_^−^ radicals via the reduction process. The holes in the valance band combine with the OH groups and oxidize the OH^−^ radicals. The formed OH^−^ and O_2_^−^ radicals act as oxidizers to degrade the molecular structure of MB dye. Ni-doping introduced Fermi levels into the In_2_O_3_ band structures and functioned as a reservoir for electrons and holes. It provides a respite for interfacial recombination and, hence, improved photodegradation. In In_2_O_3,_ a direct transition of excited electrons between the valance and conduction bands may have a faster recombination rate and lower efficacy in degrading the MB. A synergetic effect of rGO incorporation and Ni-doping led to increased charge transportation and efficient charge separation. rGO/Ni-In_2_O_3_-improved degradation may be due to the establishment of links between the Ni-doped In_2_O_3_ and rGO nanosheets. Due to enhanced electrical conductivity, as observed in the IV characteristics of our catalyst, the transfer of photo-induced electrons towards exposed surfaces of Ni- In_2_O_3_ occurs. A further decrease in the recombination losses of photogenerated electron-hole pairs is due to the movement of charges to active sites of the photocatalyst exterior via rGO. It acts like an adsorption substrate and improves the photocatalytic reaction rate.

#### 2.7.3. Photodegradation Mechanism

Upon the exposure of the photocatalyst to UV-visible light, electron-hole pairs are generated in the conduction and valence bands. The bandgap edge potentials were determined using the following Equations (4) and (5) [43]: (4)$$E_{\mathrm{cb}}=x-E_{\mathrm{ef}}-0.5E_{g}$$
(5)$$E_{\mathrm{vb}}=E_{\mathrm{cb}}+E_{g}$$

Equations (4) and (5) x show the electronegativity, E_ef_ represents the free electric energy of the material, while E_cb_ and E_vb_ denote the conduction and valance band edge positions, respectively. Upon exposure to UV-vis radiations, Ni-In_2_O_3_ nanoparticles go through a charge separation and electron shift to the conduction band, leaving a hole in the valance band. These photogenerated electrons move from the In_2_O_3_ conduction band to the conduction band of Ni due to the development of heterojunctions. Similarly, the holes from the valance band of In_2_O_3_ shift to the valance band of Ni. In the presence of the rGO matrix, these photogenerated electrons were trapped in the carbon-metal oxide heterojunction. As rGO sheets are famous for their electron conductivity, they improve the electron transfer to the photocatalyst surface and reduce electron-hole pair recombination. The reduction potential of rGO nanosheets is higher compared to O_2_/O_2_^−^ [44]. Therefore, the electrons available at the rGO surface react with oxygen to produce O_2_^−^ radicals, which further react with the water molecules to produce H_2_O_2_. This results in the reaction of H_2_O_2_ with electrons and holes on the catalyst surface and forms OH^−^. These hydroxyl radicals further react with adsorbed MB dye molecules to release harmless degraded products, i.e., H_2_O and CO_2_. The following equations may depict the above-stated degradation mechanism of MB dye in the presence of the rGO/Ni-In_2_O_3_ photocatalyst.

#### 2.7.4. Stability and Recyclability

The persistence and stability of our rGO-Ni-In_2_O_3_ samples were studied by FE-SEM and XRD analyses after using five consecutive photocatalytic cycles. At the end of each cycle, the catalyst sample was washed with DD water and heated at 90 °C for 1 h. Figure 8 shows the FE-SEM picture and XRD profile of the rGO-Ni-In_2_O_3_ sample obtained following the cyclic tests. The morphology is carefully observed and appears to be the same, with slight changes after five consecutive cycles under 1 Sun with a solar simulator. Therefore, it is concluded that the photocatalyst demonstrated excellent stability throughout the photodegradation process. Figure 8b shows the rGO-Ni-In_2_O_3_ sample XRD profiles after the five photocatalytic cycles. No obvious change in the XRD pattern is observed during the photodegradation reaction. The XRD spectrum of the used photocatalyst remains unchanged. No new phases were observed, which validates that the rGO-Ni-In_2_O_3_ sample material is a comparatively stable photocatalyst [45]. Figure 8c shows only a 3.1 percent loss in the degradability of photocatalyst after five times of use. Our experiments propose that dye removal occurred by absorption of dye on the photocatalyst surface, briefly via covalent bonding and hydrogen bonding with nitrogen in the MB dye.

## 3. Materials and Methods

To conduct the analytical studies, highly pure materials, solvents, and chemicals were purchased from Alpha Chemika (Indium (III) chloride 98%) and Alfa Aesar (nitrate hexahydrate).

The pure In_2_O_3_ was synthesized using the hydrothermal method, such as Indium (III) chloride, obtained from Alpha Chemika as an Indium precursor, and 30% ammonium hydroxide solution as a precipitating agent. For Ni-ion doping, nickel (II) nitrate hexahydrate (>98.5% purity) from Alfa Aesar was used as the Ni precursor. A total of 0.025 M Indium (III) chloride was added to 80 mL of deionized water and stirred until a clear homogeneous solution was achieved. Then, 30% ammonium hydroxide solution was added dropwise to the precursor solution while stirring for 30 min, forming precipitates. The pH of the solution remained between 6 to 10 before and after the addition of ammonium hydroxide.

For one wt% of the transition metal ion precursor, nickel (II) nitrate hexahydrate was added to the Indium precursor solution before precipitating it with the ammonium hydroxide solution. The mixture was added to a Teflon vessel, sealed into a steel container, and kept at 180 °C for 24 h in an oven. After completing the reaction, the container was kept at room temperature. In (OH)_3_, the precipitates thus formed were washed with double distilled water 3 to 4 times and dried at 80 °C. The precipitates thus obtained were further calcined at 450 C for 3 h to obtain crystalline In_2_O_3_ and Ni-In_2_O_3_ nanoparticles. Finally, 0.50 g of Ni-doped In_2_O_3_ powder was added to the reduced graphene oxide (rGO) suspension with a concentration of 1 mg/1 mL. The suspension was stirred for 30 min using an ultrasonication bath to finally obtain a well-dispersed rGO/Ni-In_2_O_3_ nanocomposite. Graphene oxide was produced using a modified Hummers method. Methylene blue was the dye used in photocatalytic experiments.

### 3.1. Characterization

#### 3.1.1. Physical Characterization of Photocatalyst

The calcined powder was characterized by a Shimadzu 6100AS X-ray diffraction (XRD) system in the 2θ range of 10–80° with Cu-Kα radiations (λ~1.5416 Å) operated at a voltage of 40 kV and a current of 30 mA. The sample’s morphology was completed using scanning electron microscopy (FEI Inspect S50) and transmission electron microscopy (TEM). UV-vis measurements of the sample were studied in the range of 200–1100 nm using the Perkin Elmer spectrophotometer. Fourier-transform spectroscopy (FTIR) was used to study the functional group characterization of the pure doped and composite samples using a Shimadzu IR Affinity-1S spectrometer. The KBr powder pellets were formed to obtain the FTIR results. The UV-visible Jenway 6850 double-beam spectrophotometer was used for the optical characterization of samples. The I-V measurements of the pellets of known thickness and the circular diameters of photocatalysts were performed using a Keithley 6517B picoammeter.

#### 3.1.2. Photoelectrochemical Measurement Methods

For electrochemical characterization, indium tin oxide (ITO) substrates with the specifications: size (l × w × t = 25 × 25 × 1.1 mm), R (resistivity = 12–15 ohms), and transmittance efficiency of more than 80%, were obtained. ITO-coated glasses loaded with our synthesized photocatalysts were prepared as working electrodes with an area of 4 cm^2^. Before use, the ITO substrates were subjected to ultrasonic cleaning for 15 min while immersed in DI water and ethanol. The sample (20 mg) was suspended in 0.5 mL of isopropanol and sonicated for 10 min. After the dispersion of the suspension was drop cast twice over the ITO and dried, a three-electrode setup containing an Ag/AgCl electrode as a reference and Pt wire as a counter electrode in 0.5 M of Na_2_SO_4_ electrolyte solution and ITO-coated glass pasted with photocatalyst was used as a working electrode using an IVIUM n STAT multichannel electrochemical analyzer. Under ambient conditions, the working electrode was irradiated in 1 Sun illumination using the LS-150 W Xe solar simulator (ABET Technologies). Cyclic voltammetry (CV) was acquired from 1 V to −1.5 V at a scan rate of 50 mV s^−1^ in the dark. Linear sweep voltammetry (LSV) was completed in the on and off light irradiation. At the same time, chronoamperometry was measured under chopped light irradiation. The electrochemical impedance spectroscopy (EIS) measurement was carried over the 30 kHz to 5 Hz frequency range in the above electrolyte, under 1 Sun illumination and dark. The electrochemical Mott–Schottky plots were recorded at a fixed frequency of 0.75 kHz without light irradiation using the following equation: (6)$$\frac{1}{C^{2}}=\pm \frac{2}{{e\varepsilon \varepsilon }_{0}A^{2}N_{d}}\;(E-E_{\mathrm{fb}}-\frac{k_{B}}{T})$$

In the above Equation (6), C stands for capacitance, ε refers to the dielectric constant of the metal oxide, and ε_0_ is the permittivity of free space. While e is the elementary charge, A is the area, N_d_ is the donor density, E is the applied potential, E_fb_ is the flat band potential, k_B_ is Boltzmann constant, and T is the absolute temperature.

#### 3.1.3. Photocatalytic Degradation of MB Dye Methods

The photocatalytic performance of our synthesized powders (pure In_2_O_3_) and doped nickel indium oxide (Ni-In_2_O_3_), and reduced graphene oxide composite (rGO/Ni-In_2_O_3_)) was studied as described below. The methylene blue dye (MB) degradation, in the presence of a known amount of synthesized photocatalyst in 1 Sun visible light under ambient conditions, was performed using the LS-150 W Xe solar simulator (ABET Technologies). MB is readily available and shows the absorption spectrum in visible light. First, a 5 ppm solution of MB dye in deionized water was prepared. Then, 100 mL of MB dye solution was added to a quartz container with 50 mg of the photocatalyst sample. It was mixed using continuous magnetic stirring for 15 min in the dark to allow the dye molecule to be adsorbed on the photocatalyst surface. Then, the solution was continuously exposed to the 1 Sunlight source for 100 min. At equal intervals of 15 min, the 8 mL of sunlight-exposed solution was collected and centrifuged. Finally, the supernatant was analyzed for UV absorption to investigate the dye removal efficiency and degradation rate. The degradation efficiency was determined using the following equation for all the photocatalyst samples (In_2_O_3_, Ni-In_2_O_3_, and rGO/Ni-In_2_O_3_).
(7)$$\left(\frac{C_{0}-C_{t}}{C_{t}}\right)\times 100=\;\%\;\mathrm{Degradation}\;$$

In Equation (7), C_t_ is the dye concentration after light exposure at a specific time interval “t”. C_0_ represents the initial dye concentration before exposure to the light.

## 4. Conclusions

Here, nanosized homogenous Ni-In_2_O_3_ powder is synthesized via the hydrothermal method, while rGO is obtained via the improved Hummer method. Photocatalytic degradation of the MB dye solution in this graphene-based catalyst showed excellent results compared with pure In_2_O_3_ or Ni-In_2_O_3_. It is noted that the rGO/Ni-In_2_O_3_ sample removed 95% of the MB dye in 100 min under visible light irradiation, whereas pure In_2_O_3_ could remove only 40%, and Ni-In_2_O_3_ removed only 50% at the same time. The enhanced removal rate can be attributed to improved light absorbance, efficient photoactive charge-carrier separation, and lower-charge recombination. The water-splitting experiment findings showed that the rGO/Ni-In_2_O_3_ photocatalyst increased the charge transfer among catalyst/electrolyte surfaces and prolonged the life of photoexcited electrons. Ni doping in In_2_O_3_ resulted in the generation of heterojunctions, which facilitate the transfer of charge from valance and conduction bands of In_2_O_3_, whereas rGO acts as a photosensitizing agent and transfers electrons to Ni-In_2_O_3_. Due to its better charge-transfer properties and large surface area, rGO has positive effects on the improvement of the photocatalytic performance of the composites. This study shows that the addition of 2D rGO sheets and Ni-In_2_O_3_ nanoparticles in composite form presents a promising photocatalyst with potential water treatment/remediation applications.

## Figures and Tables

**Figure 1 ijms-24-07950-f001:**
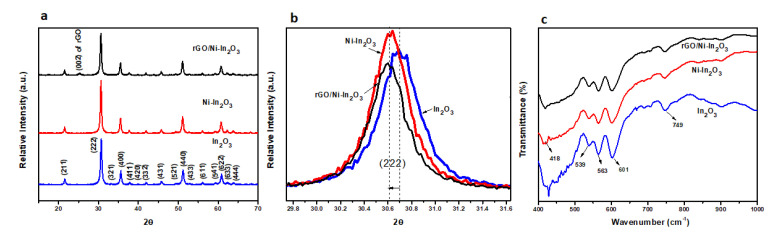
(**a**) The XRD patterns of pure In_2_O_3_, Ni-In_2_O_3_, and rGO/Ni-In_2_O_3_; (**b**) Enlarged XRD patterns of all three samples in the 2θ range of 29.6–31.6°, (**c**) FTIR profiles of pure In_2_O_3_, Ni-In_2_O_3_, and rGO/Ni-In_2_O_3_.

**Figure 2 ijms-24-07950-f002:**
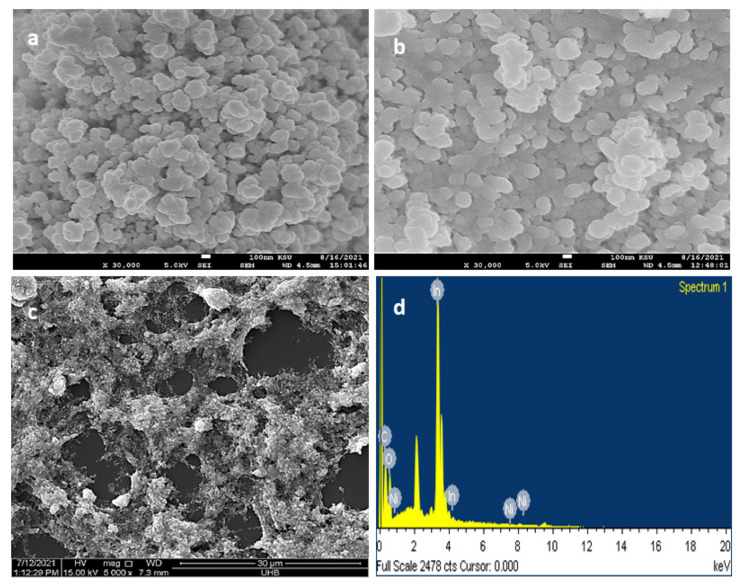
(**a**) SEM micrographs of (**a**) In_2_O_3_, (**b**) Ni-In_2_O_3_, and (**c**) rGO/Ni-In_2_O_3_. (**d**) EDX analysis of rGO/Ni-In_2_O_3_.

**Figure 3 ijms-24-07950-f003:**
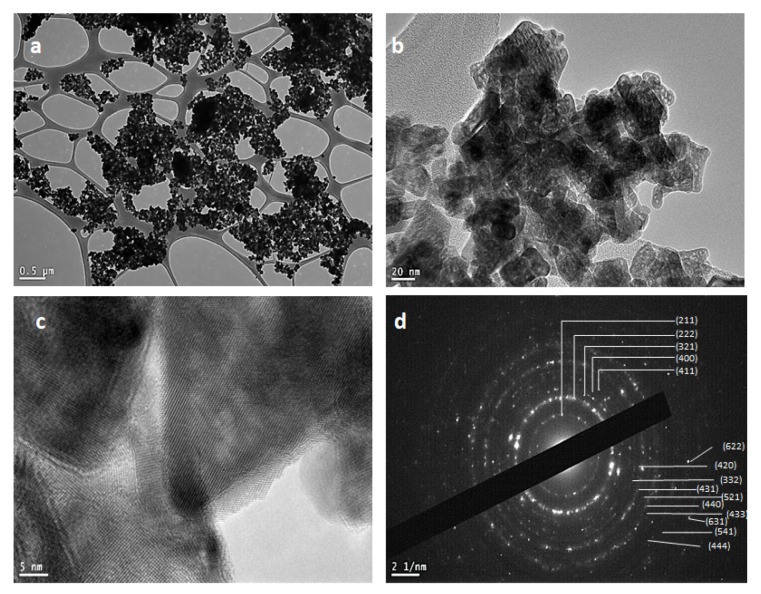
(**a**) Sample on carbon mesh (**b**) HRTEM image of rGO/Ni-In_2_O_3_; (**c**) fringe spacing of polycrystalline rGO/Ni-In_2_O_3_ nanocomposite; (**d**) SAED pattern of rGO/Ni-In_2_O_3_ micrographs.

**Figure 4 ijms-24-07950-f004:**
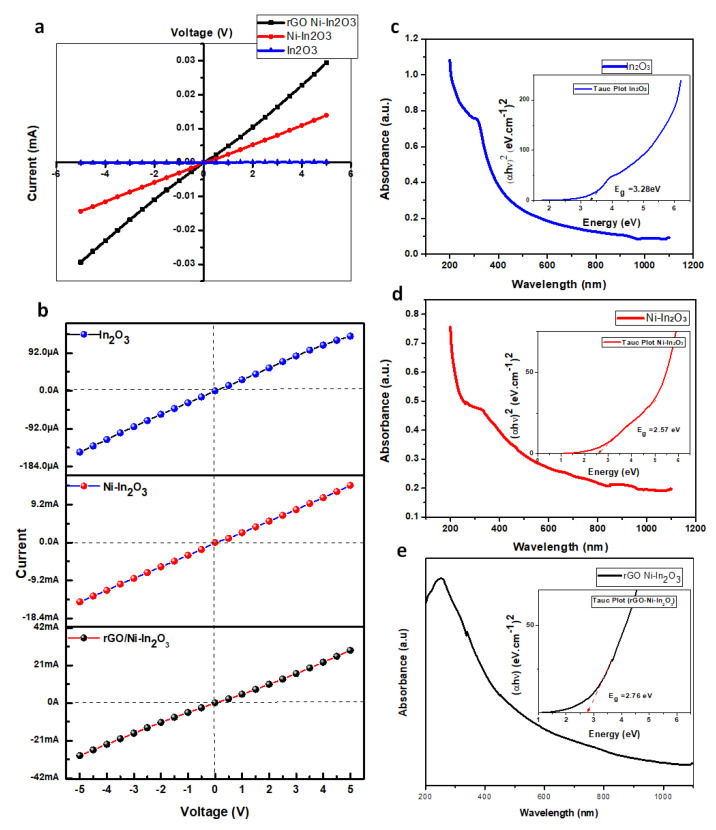
(**a**) Current and Voltage (I–V) profiles of pure In_2_O_3_, Ni-In_2_O_3_, and rGO/Ni-In_2_O_3_ nanocomposites; (**b**) comparative electrical conductivity of pure In_2_O_3_, Ni-In_2_O_3_, and rGO/Ni-In_2_O_3_ composites. (**c**–**e**) UV-vis absorption spectra of samples and the plot of (αhv)2 vs. hv (inset), based on the direct transition.

**Figure 5 ijms-24-07950-f005:**
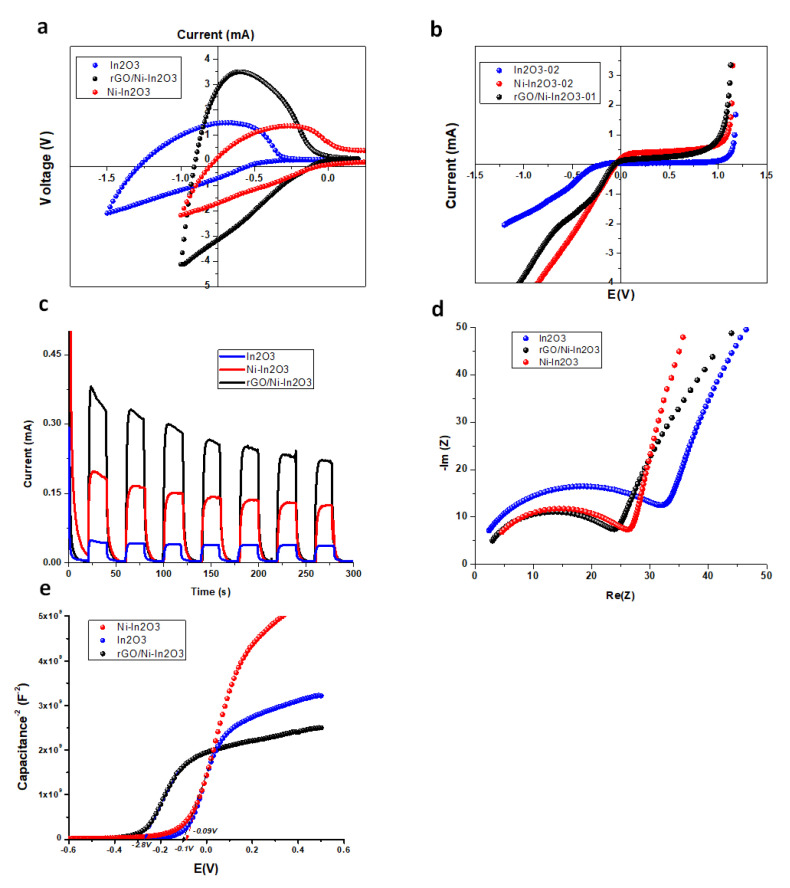
(**a**) Cyclic voltammetry in the dark; (**b**) linear sweep voltammetry data with lights on; (**c**) transient chronoamperometry; (**d**) Nyquist plots of EIS data; and (**e**) Mott–Schottky plots for In_2_O_3_, Ni-In_2_O_3,_ and the rGO/Ni-In_2_O_3_ samples.

**Figure 6 ijms-24-07950-f006:**
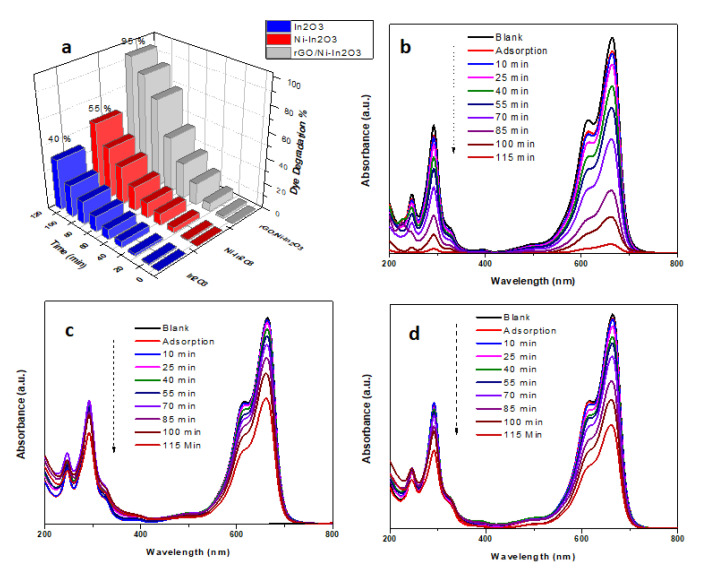
(**a**) Comparative bar chart of efficiency of all the photocatalysts. UV-vis absorption spectrum of photocatalytic degradation of MB dye at different times in the presence of the photocatalysts (**b**) rGO/Ni-In_2_O_3_ photocatalyst, (**c**) pure In_2_O_3_, and (**d**) Ni-doped In_2_O_3_.

**Figure 7 ijms-24-07950-f007:**
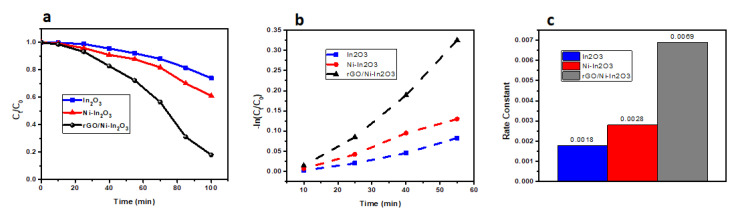
Photodegradation activity of MB dye using pure In_2_O_3_, Ni-doped In_2_O_3_, and rGO/Ni-In_2_O_3_ photocatalyst; (**a**) photodegradation rate, i.e., −ln(C_t_/C_0_) vs. time; (**b**) first-order kinetics analysis; (**c**) dye removal rate constants of all the photocatalysts.

**Figure 8 ijms-24-07950-f008:**
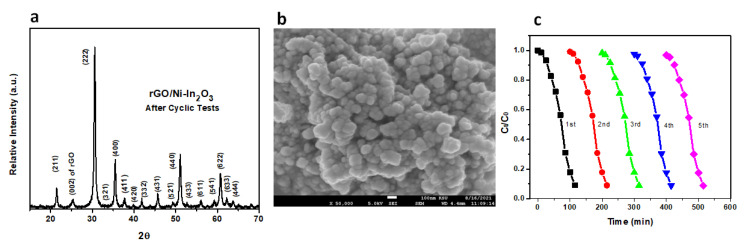
(**a**) SEM and (**b**) XRD of rGO -Ni-In_2_O_3_ after cyclic testing; (**c**) rGO-Ni-In_2_O_3_ photocatalyst’s cyclic efficiency.

## Data Availability

Available data provided with the article.

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
