# Peer review of "Ni-Doped In2O3 Nanoparticles and Their Composite with rGO for Efficient Degradation of Organic Pollutants in Wastewater under Visible Light Irradiation"

_ijms, 2023, doi:10.3390/ijms24097950_

Round 1
Reviewer 1 Report
In the manuscript, the authors adopted a facile method to construct rGO/Ni-doped In2O3 photocatalyst and investigated its activity for MB degradation under visible light. The manuscript is well-prepared, with various characterizations and data detailed, and can be considered accepted after being fully revised.
1. There are some typos in the manuscript, such as in line 88. Authors are requested to proofread the manuscript carefully.
2. The interpretation of XRD patterns needs to be more refined. In addition, in general, metal doping can lead to shifts of XRD peaks. The authors need to clarify this.
3. For the FTIR results, the author explained that the doping of Ni will cause the wavelength to increase. Why? Please give a reasonable explanation and add the necessary references.
4. The author must provide XPS of different samples to give sufficient evidence that the composite contains Ni and rGO. Otherwise, the conclusions drawn are hard to be convincing.
5. Active species capture experiments are indispensable to determine the main active species in photocatalytic degradation. At the same time, combining the energy band position can give a more reasonable photocatalytic degradation mechanism.
6. The authors need to verify the stability of the photocatalyst.
7. Suggestion: More recent papers involving photocatalytic environmental remediation can be referenced in the manuscript. (Chemical Engineering Journal, 2022, 449: 137757; Liu C, Mao S, Wang H, et al. Chemical Engineering Journal, 2022, 430: 132806; Liu C, Mao S, Shi M, et al. Journal of Hazardous Materials, 2021, 420: 126613)
Author Response
Thanks for your worthy comments. The following statements address all the observations, and replies are added with each comment. Please see the attachment.

Reviewer 2 Report
The produced catalyst appears promising as a photocatalyst. The authors carefully examined the structural and photocatalytic properties of the composite from several aspects. When reading the article, I noticed the following errors, omissions, and contradictions.
1 Corrected: „The structural and phase analysis of powdered pristine In2O3, Ni- In2O3 In2O3, and rGO/ Ni- In2O3 was performed using the X-ray diffraction (XRD) technique, as shown in Figure 1(a.)”
2 „However, the presence of Nickel and rGO cannot be observed in the XRD pattern because of low quantity or subdued signals”.- The data on nickel and graphene oxide should be included in the text.
3„In addition, the average crystallite size of calcined In2O3 and Ni- In2O3 samples is calculated using the Debye Scherer formula and found to be 21 nm and 21 nm, respectively.” „The average crystallite size of the calcined In2O3 and Ni- In2O3 samples is calculated using the Scherrer formula and found to be 21 nm and 19 nm, respectively.” -The same sentence appears twice. What is the reason for the different data?
4 „The doping of Nickel with In2O3 in the presence of rGO considerably changed the vibrations bands, such as the displacement of peaks occurring slightly towards the higher wavenumber”- The 1.b. the changes are not really visible in the diagram, they should be marked in some way.
5 The numbering and title of the chapters are not appropriate: the title of chapter 2.3 is TEM, but this will only be written about in chapter 2.4.
6„Figure. 4 (c) shows the UV–Vis absorption spectra of the pure In2O3, Ni-doped In2O3, and rGO/Ni- In2O3”-Corrected: Figure 4 (c,d,e)
7 „The bandgap calculations using Tauc plot ((αhv)2 vs. hv) in the inset of Figure 4 (c) show an increase in the bandgap with Ni- Doping might be due to a decrease in crystallite size and resulted in quantum confinement in the nanoparticle”- Incorrect sentence, other data can be read from Figures 4 c,d.
8Has the reusability of the catalysts been investigated?
9 Were TOC values measured during the decomposition of methylene blue? Knowing these data, one can only talk about mineralization.
1 In Chapter 4, the notation for oC is incorrect and the gm/ml unit is correctly mg/ml.
1 I miss the indication of the emission spectrum of the light source. A solar simulator is used as a light source. Doesn't sunlight also contain UV radiation? At the same time, they write about investigating photocatalysis in visible light. Isn't there a contradiction here? This should be clarified.
1 In the "references" section, the spelling of the titles of the referenced articles must be checked, as well as the indexing
Author Response
Thanks for your worthy comments. The following comments address all the observations, and replies are added with each comment.

Reviewer 3 Report
Upon reviewing your manuscript intitled: Ni-doped In2O3 nanoparticles and their composite with rGO for Efficient Degradation of Organic Pollutants in Wastewater under Visible Light Irradiation. I find your work interesting, but I do not believe it can be published in its current form. Therefore, some revisions must be done so that it may be published in this journal. I have the following comments and recommendations.
-The introduction of relevant background and research progress was not comprehensive enough
- The doping can improve several material properties and the choice for doping depends on several factors. With respect to this, the authors do not justify the doping concentrations used in this system. Due to the importance for this system, a reasoned explanation must be attached as part of the motivation and justification of the work.
- Include the purity of the raw materials used.
- Include the general chemical formula for the Ni-In2O3 system
- What was the Ni concentration used?
- Why the authors selected Methylene Blue (MB) for the degradation study? When there are lot of reports are available in the literature
-In Figure 1, there are several diffraction peaks that have not been identified. This must be reviewed, otherwise it is not correct information to ensure that there are no Ni or rGO peaks in the samples.
-In the rGO/Ni- In2O3 sample, there is clear evidence of an impurity phase close to 2θ = 26°. Review and identify this phase.
- How does the inclusion of Ni affect the lattice parameter of the original structure and lattice deformation? Calculate
- The crystallite sizes for the samples were calculated using Scherrer broadening formula. However, it is well known that the Scherrer peak broadening is impacted by other factors such as instrument-related broadening, residual stresses in crystals, etc. How did the authors determine or subtract these well-known parameters?
- The SEM images are of very poor quality, and figure 2c was made at a different scale. To compare samples this is very bad. Authors must provide better quality images and under the same conditions.
- Indicate in figure 3d (SAED) the rings for each plan identified and mentioned in the text. Relate to XRD results.
The quality of figure 4 is very poor, check the x axes of the UVvis spectra.
- It was mentioned that *OH free radical was the main substance for MB degradation, but the paper did not prove whether *OH, for example, analyzes by EPR, dye detection must be provided or scavengers study
- As an environmental restoration material, recycling must be considered, and some tests and analysis should be supplemented.
Author Response
Thanks for your worthy comments. The following statements address all the observations, and replies are added with each comment.

Round 2
Reviewer 1 Report
Authors addressed very well most of my comments. Paper could be published now.
Reviewer 2 Report
The authors gave appropriate answers to the questions and corrected the errors. I support the publication of the article.
Reviewer 3 Report
This version of the manuscript can be accepted for publication.